# Descriptive Prompt Paraphrasing for Target-Oriented Multimodal Sentiment Classification

**Dan Liu** [1], **Lin Li** [1], **Xiaohui Tao** [2], **Jian Cui** [1], **Qing Xie** [1]

[1] Wuhan University of Technology, Wuhan, China
[2] The University of Southern Queensland, Toowoomba, Australia
[1] {liunune, cathylilin, 318971, felixxq} @whut.edu.cn
[2] Xiaohui.Tao@unisq.edu.au

## Abstract

Target-Oriented Multimodal Sentiment Classification (TMSC) aims to perform sentiment polarity on a target jointly considering its corresponding multiple modalities including text, image, and others. Current researches mainly work on either of two types of targets in a decentralized manner. One type is entity, such as a person name, a location name, etc. and the other is aspect, such as 'food', 'service', etc. We believe that this target type based division in task modelling is not necessary because the sentiment polarity of the specific target is not governed by its type but its context. For this reason, we propose a unified model for target-oriented multimodal sentiment classification, so called UnifiedTMSC. It is prompt-based language modelling and performs well on four datasets spanning the above two target types. Specifically, we design descriptive prompt paraphrasing to reformulate TMSC task via (1) task paraphrasing, which obtains paraphrased prompts based on the task description through a paraphrasing rule, and (2) image prefix tuning, which optimizes a small continuous image vector throughout the multimodal representation space of text and images. Conducted on two entity-level multimodal datasets: Twitter-2015 and Twitter-2017, and two aspect-level multimodal datasets: Multi-ZOL and MASAD, the experimental results show the effectiveness of our UnifiedTMSC.

## 1 Introduction

With the emergence of new media and advanced technology, the forms of information released by people are quietly changing, from mono-modality to multi-modality now, such as text, images, etc (Xu and Mao, 2017). This also pushes researchers to conduct multimodal learning (Yu et al., 2016; Xu et al., 2018; Long et al., 2022). For sentiment analysis, both text and image are highly correlated with sentiment polarity. Moreover, they can complement and reinforce each other (Xu et al., 2019).

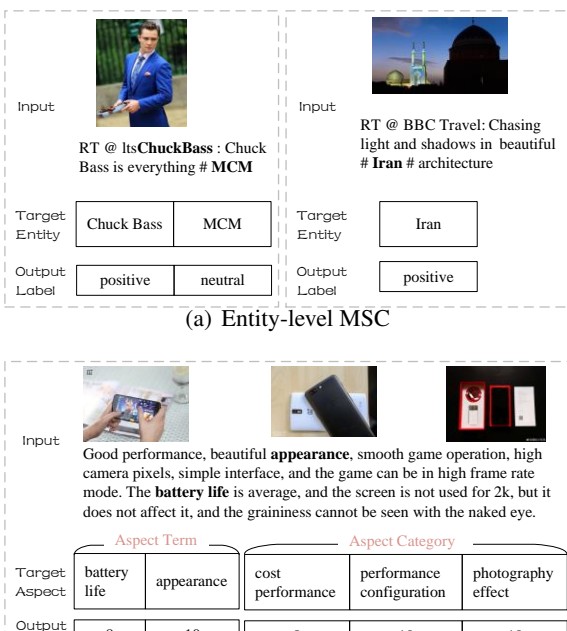

(a) Entity-level MSC

(b) Aspect-level MSC

Figure 1: Two forms of target-oriented multimodal sentiment classification.

At present, fine-grained Multimodal Sentiment Classification (MSC) includes two main tasks: entity-level MSC and aspect-level MSC, as shown in Figure 1. (1) For entity-level, the entity and its context are encoded as independent input text in some studies(Yu and Jiang, 2019; Yu et al., 2020; Wang et al., 2021; Zhao et al., 2022). Others jointly consider the encoding of an entity with its contextual learning, which is their main focus to achieve good MSC performance (Khan and Fu, 2021; Xiao et al., 2022; Yang et al., 2022). For example, Xiao et al. (2022) presented a dual stream adaptive multi-feature extraction graph convolutional network to convert an image into its caption. (2) For aspect-level, the aspect and its context are being encoded individually due to the semantics involved in an aspect itself (Xu et al., 2019; Zhang et al., 2021; Zhou et al., 2021). For example, Xu et al. (2019)

studied a multi-interactive memory network model. These TMSC works can promote human decisions by assisting users in knowing about certain targets.

As we know, an entity usually is a person name, or a location name, etc (Li et al., 2022b). Such entity target only has a specific meaning when connecting it with a specific modality content, which means it is difficult to accurately understand the entity without its context. On the contrary, an aspect itself in some degree can represent what it means even without its context. Because of this obvious difference, previous studies show their interest in differently modelling the two tasks to capture target related context (Xu et al., 2019; Yu et al., 2020; Zhang et al., 2021; Song et al., 2023). For example, in Figure 1(a), "Chuck Bass", "MCM", and "Iran" do not have any exact meaning when they are disconnected from the specific context. However, in Figure 1(b), both aspect term and aspect category have their own meanings with a hidden sentiment tendency. For the "battery life" of mobile phones, the first reaction is that the longer the battery life, the better. Despite the above differences among the TMSC tasks, from the perspective of sentiment classification, the goal of TMSC is to predict the sentiment polarity of a target no matter whether it is an entity or an aspect. Therefore, in our view, the boundary is unnecessary.

In this paper, we propose a unified TMSC model via prompt based language modelling, so called UnifiedTMSC, which is independent of the target type in TMSC. Our core is to reconstruct the two TMSC tasks through descriptive prompt paraphrasing. The prompts we design can place entity and aspect in their context, while also being close to the TMSC task description. To achieve this goal, we carry out our work from two aspects: (1) task paraphrasing. The task description is transformed into a seed prompt, and different paraphrased prompts are obtained by using our paraphrasing rule. They serve as discrete prompts for the text and fit into the Masked Language Modeling (MLM) format. And (2) image prefix tuning (Li and Liang, 2021). A segment vector is initialized for the image pretrained embedding as the prefix continuous prompt. In the subsequent multimodal continuous representation space, the image pre-trained embedding is fixed and only some segment of the initialized vector is optimized. In this way, sentiment labels are generated through the cloze-filling method.

In our extensive experiments, our UnifiedTMSC

model achieves state-of-the-art performance on two entity-level datasets: Twitter-2015 and Twitter-2017, and two aspect-level datasets: Multi-ZOL and MASAD. (1) The results of the task description based prompts are superior to those of arbitrary prompt templates. (2) On two entity-level datasets, our model improves Accuracy by 1.0%-2.8% and 1.5%-3.7%, macro-F1 by 1.5%-4.5% and 1.7%-7.5%, respectively. (3) On two aspect-level datasets, our model gains of 6.46%-8.87% and 1.66%-3.02% on Accuracy, 3.77%-5.01% and 2.19%-3.25% on macro-F1 are derived. The experimental results demonstrate the effectiveness of our UnifiedTMSC.

## 2 Related Work

### 2.1 Entity-level MSC

As a pioneer, Yu and Jiang (2019) proposed a BERT-based multimodal architecture to determine the sentiment polarity of an entity. Yu et al. (2020) introduced an entity-sensitive attention and fusion network. And Wang et al. (2021) put forward a recurrent attention network. Khan and Fu (2021) introduced an input space translation framework to construct image context from the image. Zhao et al. (2022) used the adjective-noun pairs extracted from images as the knowledge enhancement based on Yu and Jiang (2019) and Wang et al. (2021). Moreover, Yang et al. (2022) explored facial information in images to obtain visual and sentiment clues.

In the research stated above, the entity can be encoded as a distinct text input, or entity and context can be combined as the text input. Their goal is to more effectively learn the semantics related to entity sentiment.

### 2.2 Aspect-level MSC

Aspect-level multimodal classification was first proposed by Xu et al. (2019), and they introduced a multi-interactive memory network to analyze multiple correlations in multimodal data. Zhang et al. (2021) presented a multimodal fusion discriminant attention network and designed a discriminant matrix to supervise the modality fusion. Zhou et al. (2021) conducted a multimodal interaction model that learns the relationships between text, image, and target aspect through interaction layers and adversarial training. One key difference between an aspect and an entity is that the aspect has its own semantics inferred from the aspect words. Therefore, existing research usually regards the aspect

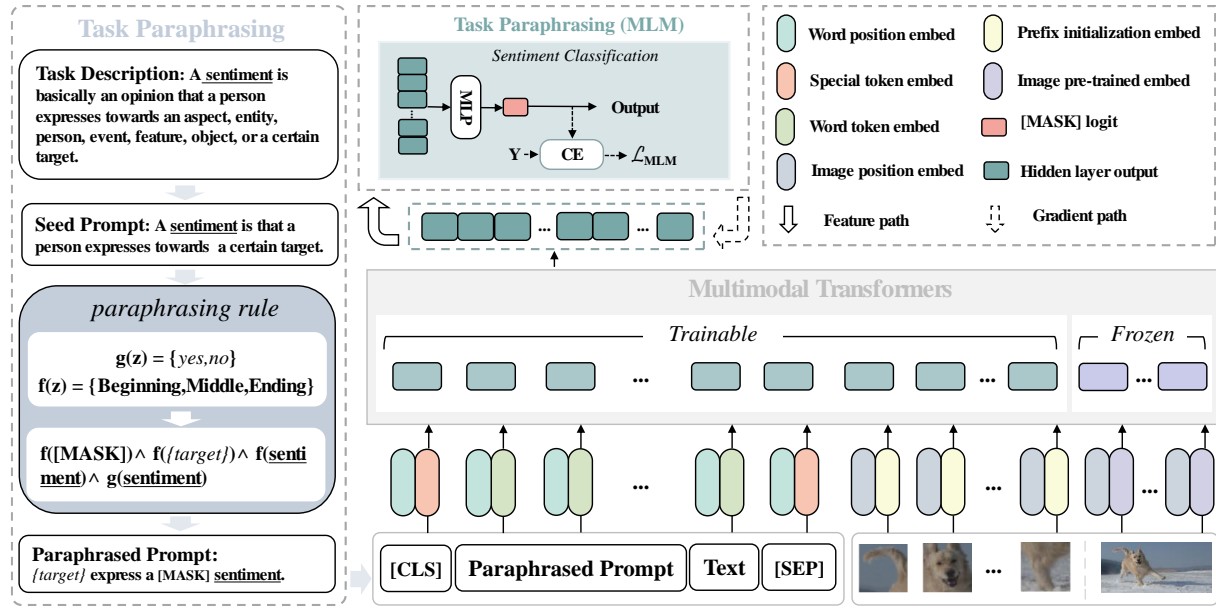

Figure 2: The model of our proposed UnifiedTMSC. Seed prompt and keyword 'sentiment' are obtained based on the task description and are converted into diverse paraphrased prompts through the paraphrasing rule (Equation 1). The paraphrased prompt and the original text input are concatenated to form the new text input which is encoded by the encoder to gain text embedding. The image is divided into multiple regions and each of the regions is initialized as a vector that is used as the prefix prompt for image pre-trained embedding. The text embedding and image embedding are undergone cross attention to gain modality fusion embedding.

itself as an additional input.

**Our focus:** In contrast to previous studies, our model can run across TMSC tasks. It combines the target (entity or aspect) and its context as a text input using task description based paraphrased prompts. We can get sentiment-related semantics about the target by providing its context in task description based prompts.

## 2.3 Prompt paraphrasing

Prompt tuning has received increasing attention recently (Radford et al., 2021; Yao et al., 2021) and has been successfully applied in many domains (Han et al., 2022; Liu et al., 2023b). For example, in the field of Question Answering (QA), Khashabi et al. (2020) reformulated many QA tasks as a text generation problem by fine-tuning seq2seq-based pre-trained models and appropriate prompts from the context and questions. For Information Extraction (IE), Chen et al. (2022) first explored the application of fixed-prompt LM Tuning in relation extraction and Lu et al. (2022) applied prompt to control the information to be extracted. In other research fields, Cui et al. (2023) used prompt learning in text input to conduct the meme mining task. Recently, the prompt has been used for the task involving fine-grained text sentiment analysis, and

the results are promising (Seoh et al., 2021; Li et al., 2021, 2022a; Gao et al., 2022; Liu et al., 2023a).

Inspired by the above studies, our unified model is through the task description based prompt paraphrasing with jointly soft and hard prompt tuning.

## 3 Method

### 3.1 Task Formulation

Given a multimodal samples $D$, for each sample $d \in D$, it contains a sentence $S$ with $n$ words $(w_1, w_2, ...w_n)$ and one or more related images $I$, as well as a target $T$ which contains $m$ words $(w_1, w_2, ...w_m)$ and is a sub-sequence of $S$ or predefined phrase. For the target $T$, it is associated with a sentiment label $Y$. In general, $Y \in \{$ *positive, neutral, negative* $\}$, and different tasks may have different sentiment labels. Our goal is to learn a target-oriented sentiment classifier that can correctly predict the sentiment label for each sample $X = (S, I, T)$.

### 3.2 Overview

As shown in Figure 2, our model consists of two modules: task paraphrasing (hard prompt) and image prefix tuning (soft prompt). For the given multimodal data $X = (S, I, T)$, we obtain paraphrased prompts $(P_1, P_2...)$ through the task description of

| Paraphrased Prompts | < Y, Position > | < T,Position > | < K, Position > | Synonym of K |
|---|---|---|---|---|
| *{target}* express a **[MASK]** sentiment. | < [MASK],M > | < *{target}*,B > | < sentiment,E > | *no* |
| The emotion of {target} is **[MASK]**. | < [MASK],E > | < {target},M > | < emotion,B > | *yes* |
| *{target}*'s emotion is **[MASK]**. | < [MASK],E > | < {target},B > | < emotion,M > | *yes* |
| The **[MASK]** feeling is *{target}*'s sentiment. | < [MASK],B > | < {target},M > | < sentiment,E > | *no* |
| The sentiment is **[MASK]** and conveyed by *{target}*. | < [MASK],M > | < {target},E > | < sentiment,B > | *no* |
| The emotion expressed by *{target}* is **[MASK]**. | < [MASK],E > | < {target},M > | < emotion,B > | *yes* |
| A **[MASK]** sentiment is expressed towards to *{target}*. | < [MASK],B > | < {target},E > | < sentiment,M > | *no* |

Table 1: Some examples of paraphrased prompts, as well as the relative positions of $Y$ (sentiment labels, replaced by [MASK] token, marked with bold), $T$ (the targets that need to recognize sentiment polarity, annotated with italics), and $K$(keywords extracted from seed prompt or replaced with synonyms, annotated with underline) in the prompts. 'Synonym of K' indicates whether to replace the keyword 'sentiment' extracted from the seed prompt with synonyms. 'B', 'M', and 'E' respectively represent the positions in the paraphrased prompt as: 'Beginning', 'Middle', and 'Ending'.

TMSC, which are used as prefixes for text input $S$ in the task paraphrasing module (the left of Figure 2). Paraphrased prompt $P_i$ contains a [MASK] token and is connected with text $S$. The text embedding $R$ is obtained through an encoder in the bottom-right of Figure 2. In the image prefix tuning module, we apply an initialization vector $V$ for pre-train image embedding $E$ as the continuous prefix. After text embedding $R$ and image embedding $(V + E)$ are added to their respective position encoding, the fusion vector is obtained through the multimodal Transformer in the middle right of Figure 2. We take the hidden layer vector $H$ and pass it through the MLM head to get the prediction score of [MASK] position for each word in the vocabulary $C$. Finally, the cross entropy loss $L_{MLM}$ of the prediction result $Output$ and the true sentiment label $Y$ is calculated.

### 3.3 Task Paraphrasing

In order to provide a specific context for a target, i.e., entity and aspect, we add the target-involved prompts to the text input. Since we change the original sentiment classification task in this way to the pre-trained MLM task, it is crucial to figure out how to develop prompts that are suitable for the original task, that is to say, the prompts should be consistent with the expression of the task description. Inspired by prompt tuning in various domains, such as the visual grounding problem (Yao et al., 2021) and visual question answering task (Liu et al., 2022), we propose the task paraphrasing module to draw paraphrased prompts as a solution to the above issue.

We get the seed prompt according to the task description composed of natural language, and take the task-related keyword $K$ ('sentiment') from the task description. The seed prompt is transformed through the paraphrasing rule, and guides the generation of paraphrased prompts that are close to the original task description. Our paraphrasing rule can be formalized in the following form:

$$f(Y) \wedge f(T) \wedge f(K) \wedge g(K) \qquad (1)$$

where the function $f$ represents the relative position in the paraphrased prompts and the substitution of synonyms for a keyword ('sentiment') is illustrated by the function $g$. $f(Y), f(T), f(K) \in \{B, M, E\}$ (meaning 'Beginning', 'Middle', 'Ending'), which respectively stands for the relative position of the sentiment label, the target entity, and the keyword derived from the task description. $g(K) \in \{yes,no\}$ means whether to replace synonyms of keywords $K$. Moreover, synonyms are synonymous explanations for the keyword 'sentiment' in the dictionary. For example, in the Bing dictionary, the synonyms for 'sentiment' include 'emotion', 'feeling', 'opinion', etc.

The task paraphrasing module is shown in the left part of Figure 2. Generally speaking, based on the relative position combination of $Y$, $K$, and $T$, one seed prompt can be paraphrased to gain multiple candidate prompts. If we replace the keyword with different synonyms, we will receive more paraphrased prompts. Some examples of paraphrased prompts $P_i$ are listed in Table 1. When paraphrased prompt $P_i$ is in the training phase, the position of $Y$ is replaced by [MASK] token which is the prediction object. Finally, the $P_i$ and text $S$ are concatenated to obtain a new text input:

$$S^{new} = [CLS] \, P_i \, S \, [SEP] \qquad (2)$$

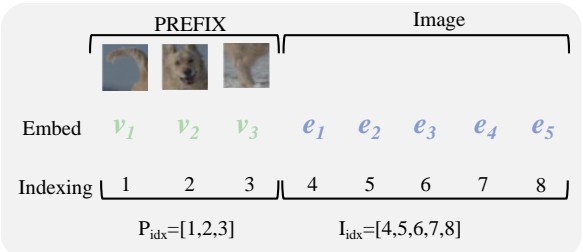

Figure 3: An example of image prefix tuning. $v_{P_{idx}}$ is random initialization of the prefix sequence and $e_{I_{idx}}$ is the image pre-trained embedding.

## 3.4 Image Prefix Tuning

Discrete prompts, i.e., hard prompts, in the text are natural and easy to understand. For images, directly applying discrete prompts cannot ensure alignment between text based prompts and images since there are modality gaps between different modalities. Therefore, finding suitable prompts for images is crucial. This module mainly focuses on how to add prompts to images to facilitate a better modality fusion of images and text.

Inspired by (Tsimpoukelli et al., 2021) and (Li and Liang, 2021), we introduce a continuous vector as a prefix, i.e., soft prompt, to image pre-trained embedding, as shown in Figure 3. We first segment the image into $r$ regions, and then initialize a vector $v_i$ ($i \in \{1, 2, ...r\}$) for each region to form the prefix embedding $V = \{v_1, v_2, ...v_r\}$ and $V \in \mathbb{R}^{r \times 2048}$. Here, $P_{idx}$ denotes the indices of the prefix sequence. $|P_{idx}|$ denotes the length of the prefix and $|I_{idx}|$ indicates the length of the image pre-trained embedding. In $E = \{e_1, e_2, e_3, e_4, e_5\}$, $e_i \in \mathbb{R}^{2048}$ refers to the image embedding of the $i$-th region. The prefix embedding $V = \{v_1, v_2, v_3\}$ and $V \in \mathbb{R}^{3 \times 2048}$. If there are multiple images corresponding to the text, we initialize a soft prompt for each image and their averaged vector is used as the final prefix prompt for the images. $V$ and $E$ are concatenated to obtain new image embedding $E^{new}$.

$$E^{new} = V \oplus E \qquad (3)$$

$$E^{new}_{(i)} = \begin{cases} v_i & i \in P_{idx} \\ e_{(i-|P_{idx}|)} & otherwise \end{cases} \qquad (4)$$

$E^{new}$ and $S^{new}$ are image input and text input respectively, and the subsequent encoding process is carried out together.

| | Twitter-2015 | Twitter-2017 | Multi-ZOL | MASAD |
|---|---|---|---|---|
| | Text-Image | Text-Image | Text-Image | Text-Image |
| Train | 3179 | 3562 | 22775 | 26292 |
| Dev | 1122 | 1176 | 2847 | 2925 |
| Test | 1037 | 1234 | 2847 | 8850 |
| Total | 5338 | 5972 | 28469 | 38532 |

Table 2: Statistics of four TMSC datasets.

## 3.5 Multimodal Transformer With MLM

**Training.** Given a triplet $(S, I, T, Y)$, after the task paraphrasing module and image prefix tuning module, the updated text input $S^{new}$ and image embeddings $E^{new}$ are obtained. Text encoder encodes $S^{new}$ to gain text embedding $R$. $R$ and the image embedding $E^{new}$ undergo cross-attention through a Transformer to obtain the fusion embedding $H$. In this multimodal Transformer, the image pre-trained embedding $E$ is fixed and not updated, and only $V$ is updated. Fusion embedding $H$ passes through an MLP to obtain prediction scores $Logit$:

$$Logit = MLP_{MLM}(H) \qquad (5)$$

The word with the highest prediction score is the prediction result:

$$Output = argmax(Logit) \qquad (6)$$

Finally, the predicted result and the true sentiment label $Y$ are calculated in the cross-entropy loss to optimize our model.

$$L_{MLM} = -\log(\frac{\exp(Logit_i)}{\sum_{j=0}^{|C|} \exp(Logit_j)}) \qquad (7)$$

where $i$ is the true sentiment label number and $C$ is the size of the language model vocabulary.

**Inference.** In the inference stage, given a triplet $(S, I, T)$, the sentiment polarity $Y'$ of $T$ is determined via the triplet.

$$Y' = argmax(Logit_{[MASK]}) \qquad (8)$$

After obtaining the fusion embedding of text and images, we take the argmax of the logit of the [MASK] position to obtain the final prediction result. Finally, for non-label words generated by the model, the answer engineering is used to map them to sentiment labels.

## 4 Experiment

### 4.1 Experimental Setup

**Datasets.** According to Zhou et al. (2021), there are four datasets for target-oriented multimodal

sentiment classification, including two entity-level datasets: Twitter-2015 (Zhang et al., 2018; Yu and Jiang, 2019),Twitter-2017 (Lu et al., 2018; Yu and Jiang, 2019) and two aspect-level datasets: Multi-ZOL (Xu et al., 2019), MASDA (Zhou et al., 2021) The statistics information of these four datasets are shown in Table 2, and their details about the data partitioning and label types are presented in A.2.

**Evaluation Metrics.** To fairly compare with state-of-the-art approaches, our UnifiedTMSC is evaluated across two TMSC tasks and adopts the Accuracy (Acc) and Macro-F1 score (F1), following Yu and Jiang (2019) and Ling et al. (2022).

**Implementation Details.** For the Twitter-2015, Twitter-2017, and MASAD, the batch size is set to 16 and the epochs are set to 6. The text encoder used is BERT-base-uncased (Devlin et al., 2019). For the Multi-ZOL dataset, in order to ensure fairness in the experiment, we employ BERT-base-chinese (Devlin et al., 2019) as the text encoder, the batch size is 8, and the epochs as 6. Moreover, multilingual pre-trained models can also be used as our text encoders.

For all datasets, we apply Resnet-50 (He et al., 2016) as the image encoder to get the image prefix embedding, and the max length of the new text input $S^{new}$ (in Eq. (2)) is 96. The model learning rate is set as 1e-5 and the dropout rate is 1e-2. Four layer Transformers (Vaswani et al., 2017) are aimed to perform cross attention between different modalities and the pre-training parameters are not loaded. Four NVIDIA TITAN Xp GPUs, each with 12GB of memory, are employed in our experiments which are done on a CentOS computer. The deep learning framework is Pytorch, and AdamW is used as the optimizer.

## 4.2 Compared Baselines

Because previous work has separated entity-level MSC and aspect-level MSC, the baseline models for each task are different. For the Twitter-2015 and Twitter-2017 datasets, we compare six baselines including TomBERT (IJCAI, Yu and Jiang (2019)), SaliencyBERT (PRCV, Wang et al. (2021)), CapTrBERT (ACM Multimedia, Khan and Fu (2021)), JML-MASC (EMNLP, Ju et al. (2021)), VLP-MABSA (ACL, Ling et al. (2022)), FITE-DE-Large (EMNLP, Yang et al. (2022)). For the Multi-ZOL and MASAD datasets, our model is compared with MIMN (AAAI, Xu et al. (2019)), ModalNet (WWW, Zhang et al. (2021)), MMAP

| Method | Twitter-2015 | | | Twitter-2017 | | |
|---|---|---|---|---|---|---|
| | Acc | F1 | P-Value | Acc | F1 | P-Value |
| TomBERT | 77.2 | 71.8 | 0.0194 | 70.5 | 68.0 | 0.0156 |
| SaliencyBERT | 77.0 | 72.4 | 0.0109 | 69.7 | 67.2 | 0.0156 |
| CapTrBERT | 78.0 | 73.3 | 0.0120 | 72.3 | 70.2 | 0.0134 |
| JML-MASC | 78.7 | - | - | 72.7 | - | - |
| VLP-MABSA | 78.6 | 73.8 | 0.0131 | 73.8 | 71.8 | 0.0111 |
| FITE-DE-Large | 78.8 | 74.8 | 0.0048 | 73.9 | 73.0 | 0.0016 |
| UnifiedTMSC | **79.8** | **76.3** | - | **75.4** | **74.7** | - |

Table 3: The experimental results on the multimodal entity-level datasets: Twitter-2015 and Twitter-2017. The results presented in the table are the average of different prompt results.

| Method | Multi-ZOL | | | MASAD | | |
|---|---|---|---|---|---|---|
| | Acc | F1 | P-Value | Acc | F1 | P-Value |
| MIMN | 61.59 | 60.51 | 0.0316 | 94.52 | 93.55 | 0.0019 |
| ModalNet | 62.71 | 60.94 | 0.0266 | - | - | - |
| MMAP | - | - | - | 95.29 | 94.61 | 0.0049 |
| CLIP | 64.00 | 59.70 | 0.0092 | 95.88 | 94.21 | 0.0079 |
| UnifiedTMSC | **70.46** | **64.71** | - | **97.54** | **96.80** | - |

Table 4: The experimental results on the multimodal aspect-level datasets: Multi-ZOL and MASAD.

(Neurocomputing, Zhou et al. (2021)) and CLIP (INT J COMPUT VISION, Zhou et al. (2022)). The detailed introduction of all the baseline models mentioned above is in Section A.1.

## 4.3 Experimental Results and Analysis

### 4.3.1 Overall Performance

The experimental results on multimodal entity-level and aspect-level datasets are presented in Table 3 and Table 4 respectively. The best results on each metric are marked in bold and the second best results are highlighted with an underline.

**Multimodal Entity-level Datasets Results.** As reported in Table 3, compared with the baselines, our UnifiedTMSC makes significant improvements in entity-level MSC. On the Twitter-2015 dataset, our improvement is approximately 1.0% on Accuracy and 1.5% on the macro-F1 compared to the FITE-DE-Large. On the Twitter-2017 dataset, we achieve improvements over multimodal baseline VLP-MABSA on the Accuracy by 1.5% and 1.7% on the macro-F1, which indicates that using prompt tuning to fuse entity and context can achieve a better sentiment-related semantic understanding of an entity, resulting in better classification results.

**Multimodal Aspect-level Datasets Results.** Our UnifiedTMSC performs best in two datasets among all multimodal baselines, as listed in Table 4. This demonstrates the effectiveness of our proposed unified model based on prompt paraphrasing. Especially on the Multi-ZOL dataset, our UnifiedTMSC outperforms the ModalNet on the Accuracy by 7.75% and 3.77% on the macro-F1. On

| Method | | Food | | Buildings | | Goods | | Human | | Scenery | |
|---|---|---|---|---|---|---|---|---|---|---|---|
| | | Acc | F1 | Acc | F1 | Acc | F1 | Acc | F1 | Acc | F1 |
| MIMN | AAAI 2019 | 94.72 | 91.39 | 96.26 | 95.80 | 95.93 | 95.87 | 92.54 | 92.31 | 93.17 | 92.38 |
| MMAP | Neucomp 2021 | 95.75 | 92.89 | 96.86 | 96.85 | 96.55 | 96.44 | 92.74 | 92.74 | 94.57 | 94.15 |
| CLIP | IJCV 2022 | 96.15 | 93.05 | 95.84 | 94.27 | 97.36 | 96.88 | 94.20 | 93.24 | **95.84** | 93.59 |
| UnifiedTMSC | | **98.00** | **96.02** | **98.26** | **98.25** | **98.08** | **98.01** | **97.82** | **96.49** | 95.56 | **95.24** |

Table 5: The experimental results of each domain in the MASAD dataset. Because the accuracy and Macro F1 score in the domain of animal and plant reach 99.07%-99.22% due to the duplicated samples, they will not be displayed here.

the MASAD dataset, compared to the MMAP, ours improve performance by about 2.25% on Accuracy and 2.19% on the macro-F1.

Specifically, there are multiple domains in the MASAD dataset, we conduct experiments on each of them, and the experimental results are shown in Table 5. It is clear from the results that our model has achieved large improvements in each domain.

Finally, t-tests are conducted to demonstrate the effectiveness of UnifiedTMSC. From the P-value of other models in Table 3 and Table 4, it can be found that all P-values are less than 0.05. This shows a significant difference in statistics between UnifiedTMSC and other models.

After comparing and analyzing the experimental results, we can summarize the following two points from our prompt tuning:

I. Our prompt paraphrasing method delivers the target's context and fits the TMSC job effectively, and it produces good results, demonstrating the efficacy of our unified model.

II. Utilizing the target as a separate input has worse results than taking the target and context together as text input. This shows that contextual information affects the target's semantics, and a contextual content that is appropriate for the task will result in a well understanding of a target with semantics.

### 4.3.2 The Effect of Prompt Designs

For the Twitter-2015 and Twitter-2017 datasets, we select three paraphrased prompts from Table 1 to conduct the experiments. The three selected paraphrased prompts are as follows:
- $P_1$: *{target}* express a **[MASK]** sentiment.
- $P_2$: The emotion of *{target}* is **[MASK]**.
- $P_3$: A **[MASK]** sentiment is expressed towards to *{target}*.

In addition, to verify the performance of the task paraphrasing module. We design three arbitrary prompt templates and compare them with the above

| | | Twitter-2015 | | Twitter-2017 | |
|---|---|---|---|---|---|
| | | Acc | F1 | Acc | F1 |
| paraphrased prompts | $P_1$ | 80.2 | **77.6** | **76.9** | **76.7** |
| | $P_2$ | **81.1** | 77.1 | 76.6 | 75.9 |
| | $P_3$ | 79.8 | 77.5 | 76.2 | 75.9 |
| arbitrary prompts | $P_1'$ | 79.1 | 75.5 | 73.8 | 73.7 |
| | $P_2'$ | 79.0 | 75.1 | 74.3 | 73.6 |
| | $P_3'$ | 79.4 | 75.1 | 74.5 | 72.9 |

Table 6: The experimental results of different prompts. The results of paraphrased prompts are superior to the arbitrary prompts, and $P_1'$, $P_2'$ and $P_3'$ are comparable. In the experiments, one of the paraphrased prompts can be selected for experimentation.

three paraphrased prompts. The three arbitrary prompts are as follows:
- $P_1'$: I feel the *{target}* is **[MASK]**.
- $P_2'$: The *{target}* made me feel **[MASK]**.
- $P_3'$: I **[MASK]** the *{target}*.

where *{target}* is the entity that needs to determine sentiment polarity, and **[MASK]** represents the masked word, i.e. sentiment label. The masked word in the $P_1'$, $P_2'$, $P_3'$ is *{good, ok, bad}, {good, indifferent, bad}* and *{love, dislike, hate}* respectively. After the masked words are generated, we perform answer engineering to map the predicted results to the sentiment polarity set, that is, the probabilities of these predicted words are made to be equal to the probabilities of being $Positive$, $Neutral$, and $Negative$.

The results of several different prompts are shown in Table 6. Through analysis and comparison, we can obtain the following summaries:

I. Our paraphrased prompts created by using the task description are much superior to arbitrary prompt templates, demonstrating the value of our task paraphrasing module in producing paraphrased prompts that are appropriate for the original sentiment classification task. In addition, the performance of different paraphrased prompts is comparable, and in subsequent experiments, anyone can be selected for training and inference.

| | (a) | (b) | (c) | (d) |
|---|---|---|---|---|
| Image | | | | |
| Text | (a) Fan Throws Water Bottle at Justin Bieber After He Says He Doesn ' t Know the . . . | (b) Kim Kardashian goes all out for Kanye West ' s 40 th birthday in the Bahamas | (c) RT @ TrumpDoral : Congratulations to the the new # MissUniverse , Miss Colombia , Paulina Vega ! | (d) Nba - The Cavs are Shocked Draymond Green Keeps Getting Away With Kic - |
| (Target,Label) | (Justin Biebe, Negative) | (Kim Kardashian, Positive) (Kanye West, Positive) (Bahamas, Neutral) | (Miss Colombia, Positive) (Paulina Vega, Positive) | (Nba, Neutral) (Cavs, Neutral) (Draymond Green, Negative) |
| w/o Paraphrased Prompt | (Justin Biebe, Neutral) × | (Kim Kardashian, Neutral) × (Kanye West, Neutral) × (Bahamas, Neutral) ✓ | (Miss Colombia, Neutral) × (Paulina Vega, Neutral) × | (Nba, Neutral) ✓ (Cavs, Negative) × (Draymond Green, Positive) × |
| w/o Image Prefix | (Justin Biebe, Positive) × | (Kim Kardashian, Positive) ✓ (Kanye West, Neutral) × (Bahamas, Neutral) ✓ | (Miss Colombia, Neutral) × (Paulina Vega, Positive) ✓ | (Nba, Neutral) ✓ (Cavs, Negative) × (Draymond Green, Negative) ✓ |
| UnifiedTMSC | (Justin Biebe, Negative) ✓ | (Kim Kardashian, Positive) ✓ (Kanye West, Positive) ✓ (Bahamas, Neutral) ✓ | (Miss Colombia, Positive) ✓ (Paulina Vega, Positive) ✓ | (Nba, Neutral) ✓ (Cavs, Neutral) ✓ (Draymond Green, Negative) ✓ |

Figure 4: Case study on four test samples. Red font indicates correctly predicted labels.

| | Twitter-2015 | | Twitter-2017 | |
|---|---|---|---|---|
| | Acc | F1 | Acc | F1 |
| UnifiedTMSC | **79.8** | **76.3** | **75.4** | **74.7** |
| w/o Paraphrased Prompts | 75.7 | 72.0 | 68 | 65.9 |
| $P_1$ + w/o Image Prefix | 78.9 | 75.2 | 74.1 | 73.7 |
| $P_2$ + w/o Image Prefix | 78.6 | 75.7 | 74.4 | 73.8 |
| $P_3$ + w/o Image Prefix | 78.5 | 75.7 | 74.3 | 73.9 |
| AVG w/o Image Prefix | 78.7 | 75.5 | 74.3 | 73.8 |

Table 7: Ablation study of our UnifiedTMSC model.

II. The position of [MASK] in the paraphrased prompts can also have an impact on the experimental results. In our case, the effect is best when the relative position of [MASK] is "Middle" rather than "Beginning" or "Ending". Therefore, when meeting a new task, the position of [MASK] may be a factor to affect task performance.

## 4.4 Ablation Study

To further investigate the effects of paraphrased prompts and image prefix, because entity-level MSC is more challenging than aspect-level MSC, we conduct ablation analysis on the multimodal entity-level datasets: Twitter-2015 and Twitter-2017. The results of the ablation experiments are shown in Table 7.

**Paraphrased Prompts.** $P_i$ is omitted from Eq. (2) and we just add the image prefix $V$ in Eq. (3) for the experiment in order to examine the effects of the paraphrased prompts. The linear classification layer uses the fusion vector derived by multimodal Transformers as input to estimate the sentiment la-

bel of the target. The results are shown in Table 7. The absence of paraphrased prompts has resulted in a considerable performance decrease. On the Twitter-2015 dataset, the Accuracy and macro-F1 are dropped by approximately 4.1% and 4.3%, while on the Twitter-2017 dataset, the Accuracy declines by about 7.4% and the macro-F1 drops by about 8.8%. This demonstrates how using text paraphrased prompts can provide an entity with its task-related semantics.

**Image Prefix.** We only use the paraphrased prompt $P_i$ in Eq. (2) without applying the image prefix prompt $V$ in Eq. (3) to study the importance of the image prefix. For text input, $P_1$, $P_2$, and $P_3$ are proceeded for the ablation study. From Table 7, it can be seen that the performance has dropped after taking out the image prefix $V$ (in Eq. (3)). The Accuracy decreases by 1.1%, and the macro-F1 drops by 0.8%, according to the average results for the Twitter-2015 dataset. On the Twitter-2017 dataset, the Accuracy and macro-F1 decline by roughly 1.0%. This illustrates the effectiveness of the image prefix. Moreover, the ablation study also shows that different paraphrased prompts have varied outcomes, demonstrating the language models' sensitivity to prompts.

## 4.5 Case Study

In our case study, the compared methods are only image prefix (denoted by w/o Paraphrased Prompt), only text prompt (denoted by w/o Image Prefix),

and our UnifiedTMSC model with soft and hard prompts. We apply $P_1$ for both w/o Image Prefix and UnifiedTMSC.

As shown in Figure 4, for example (a), when there is no paraphrased prompt, the result obtained from text and image information is *Neutral*. When there is no image prefix, the image pre-trained embedding dominates the prediction results as *Positive*. Both of these two prediction results do not match the correct sentiment label *Negative*. For example (b), there are multiple targets that require sentiment classification. The sentiment labels predicted for place name *Bahamas* are *Neutral*, and adding appropriate prompts to both image and text can be predicted correctly. For examples (c) and (d), they are similar to example (b).

These four samples further confirm the usefulness of our unified model. It can assign specific sentiment-related semantics to an entity via applying a paraphrased prompt. And prefix tuning of images can obtain better task-specific image embedding than image pre-trained embedding.

## 5 Conclusion and Future Work

There are currently two formats for target-oriented multimodal sentiment classification: entity-level and aspect-level. Our analysis shows that this barrier is superfluous. By incorporating paraphrased prompt and prefix vector into the multimodal input, the proposed model, i.e., UnifiedTMSC, unifies the two types of TMSC tasks. We conduct experiments on four datasets, and the results demonstrate the superiority and efficacy of our UnifiedTMSC.

Our ongoing effort will primarily concentrate on two issues. One is to investigate how to design a paraphrasing rule to automatically generate paraphrased prompts without depending on human labor. The other is to investigate the generalizability of our model to see whether it can be used in other multimodal studies. In addition, we notice that the auto-regressive model XLNet can alleviate the problem of generating non-label words, and our future work will consider this.

## Limitations

Our model has three limitations. The first one is that the design of paraphrased prompts relies on human experience. Although our paraphrasing rule is designed based on the relative position and synonym substitution, manual experience is still required to obtain paraphrased prompts that comply

with grammar rules, and paraphrased prompts that comply with grammar rules may not necessarily be the best. The second limitation is that more attempts are needed to conduct experiments on multilingual pre-trained models. Furthermore, the last is that we have not explored whether our model can be extended to other multimodal research fields, which will be our future research direction.

## Acknowledgments

This work is partially supported by NSFC, China (No.62276196).

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

# A  Appendix

## A.1  Baselines

(1) TomBERT (IJCAI, Yu and Jiang (2019)), which employ BERT for the inter-modal interactions and Target-Image matching layer to obtain a target-sensitive visual.

(2) SaliencyBERT (PRCV, Wang et al. (2021)), which design a recurrent attention mechanism to capture the inter-modality dynamics.

(3) CapTrBERT (ACM Multimedia, Khan and Fu (2021)), a BERT-based model translating the image into caption and fusing the caption and input text-entity pair.

(4) JML-MASC (EMNLP, Ju et al. (2021)), which is a multi-task learning method with the cross-modal relation detection.

(5) VLP-MABSA (ACL, Ling et al. (2022)), a task-specific pre-training vision-language model.

(6) FITE-DE-Large (EMNLP, Yang et al. (2022)), introducing a FITE method to focus on capturing emotional cues through facial expressions.

(7) MIMN (AAAI, Xu et al. (2019)), using memory network to model multimodal data and learn the interactive influences in cross-modality and self-modality.

(8) MMAP (Neurocomputing, Zhou et al. (2021)), learning the interaction between text and image, text and aspect, and image and aspect through three interactive mechanisms.

(9) ModalNet (WWW, Zhang et al. (2021)), designing a discriminant matrix to supervise the fusion of inter-modal information.

(10) CLIP (INT J COMPUT VISION, Zhou et al. (2022)), a multimodal pre-training model, converts image classification tasks into image-text matching tasks using comparative learning.

## A.2  Dataset Details

(1) **Twitter-2015 and Twitter-2017.** The two Twitter datasets include user tweets released during 2014-2015 and 2016-2017, collected by Zhang et al. (2018) and Lu et al. (2018). Since the two publicly available multimodal datasets Twitter-2015 and Twitter-2017 only provide annotated targets in each tweet, Yu and Jiang (2019) ask three domain experts to annotate the sentiment towards each target, and take the majority label among the three annotators as the gold label. These two datasets contain multimodal tweets and annotated mentioned entities in the text, as well as the sentiment polarity of each entity, including *positive, neutral, and negative*. Each multimodal tweet consists of a text and a corresponding image. We follow their official splitting for training, validation and testing. The statistics of Twitter-2015 and Twitter-2017 are in Table 8.

(2) **Multi-ZOL.** This dataset is collected by Xu et al. (2019) from the Chinese website ZOL.com. The website consists of 40 large channels, including news, shopping malls, hardware, mobile phones, and more. They searched from pages 1 to 50 in the mobile phones channel. For each phone, only the reviews from the first 20 pages were crawled. The Multi-ZOL dataset contains 5288 multimodal reviews and each multimodal review contains a textual content, an image set, and at least one but no more than six aspects. The six aspects are cost performance, performance configuration, battery life, appearance, photography effect, and screen. For each aspect, the review has an integer sentiment score from 1 to 10, which is regarded as the sentiment label in our experiment. Actually, we convert digital sentiment labels into Chinese characters, such as '10' to 'ten'. Combining each aspect with the multimodal review, 28469 pairs of aspect review samples can be obtained. We randomly partition the dataset into training (80%), development (10%), and testing (10%) sets. The statistical information after dividing the dataset is shown in Table 9.

(3) **MASAD.** This dataset was collected and published by Zhou et al. (2021) based on the publicly available Visual Sentiment Ontology (VSO) dataset (Borth et al., 2013) and Multi-

|  | Twitter-2015 | | | | Twitter-2017 | | | |
|  | #Positive | #Neutral | #Negative | Total | #Positive | #Neutral | #Negative | Total |
|---|---|---|---|---|---|---|---|---|
| Train | 928 | 1883 | 368 | 3179 | 1508 | 416 | 1638 | 3562 |
| Dev | 303 | 679 | 149 | 1122 | 515 | 144 | 517 | 1176 |
| Test | 317 | 607 | 113 | 1037 | 493 | 168 | 573 | 1234 |

Table 8: The statistics of Twitter-2015 and Twitter-2017 datasets.

|  | #1 | #2 | #3 | #4 | #5 | #6 | #7 | #8 | #9 | #10 | Total | Max #Image | Min #Image |
|---|---|---|---|---|---|---|---|---|---|---|---|---|---|
| Train | 6 | 1163 | 6 | 1190 | 3 | 3509 | 0 | 6952 | 0 | 9948 | 22775 | 111 | 1 |
| Dev | 0 | 139 | 0 | 150 | 0 | 433 | 0 | 872 | 0 | 1253 | 2847 | 111 | 1 |
| Test | 0 | 132 | 0 | 151 | 0 | 413 | 0 | 871 | 0 | 1280 | 2847 | 72 | 1 |

Table 9: The statistics of Multi-ZOL dataset after partitioning. 'Max #Image' refers to the maximum number of image corresponding to the review, while 'Min #Image' is the minimum quantity.

|  | Train | | | Dev | | | Test | | |
|  | #Positive | #Negative | Total | #Positive | #Negative | Total | #Positive | #Negative | Total |
|---|---|---|---|---|---|---|---|---|---|
| Food | 2122 | 392 | 2514 | 239 | 41 | 280 | 592 | 109 | 701 |
| Buildings | 969 | 874 | 1843 | 109 | 96 | 205 | 273 | 245 | 518 |
| Goods | 2398 | 1512 | 3910 | 273 | 162 | 435 | 743 | 512 | 1255 |
| Animal | 2728 | 1979 | 4707 | 295 | 229 | 524 | 1126 | 670 | 1796 |
| Human | 1805 | 1648 | 3453 | 194 | 190 | 384 | 503 | 464 | 967 |
| Plant | 2542 | 2341 | 4883 | 277 | 266 | 543 | 1269 | 947 | 2216 |
| Scenery | 3222 | 1760 | 4982 | 378 | 176 | 554 | 907 | 490 | 1397 |
| Total | 15786 | 10506 | 26292 | 1765 | 1160 | 2925 | 5413 | 3437 | 8850 |

Table 10: The statistical information of MASAD datasets. The seven domains are Food, Buildings, Goods, Animal, Human, Plant, Scenery.

lingual Visual Sentiment Ontology (MVSO) dataset (Jou et al., 2015). Zhou et al. (2021) selected the samples from a partial VSO dataset (approximately 120k samples) that can express significant sentiments (about 38k samples) and categorized them into 7 domains, resulting in the MASAD dataset. The seven domains of the MASAD are food, buildings, goods, animal, human, plant, and scenery. Each domain encompasses multiple aspects, such as the animal domain, including cat, dog, horse, and so on. According to our statistics, there are a total of 57 predefined aspects. This dataset only includes training and testing sets, both containing *positive* and *negative* samples. We partition each domain in the training set into a new training set and a validation set in a 9:1 ratio, keeping the original testing set unchanged. The statistics of MASAD are in Table 10.

In addition, there are instances of duplicate data in the testing and training sets in this dataset. In the domain of animal and plant, according to our statistics, about 81.6% and 62.2% of the data in the testing set has appeared in the training set. In order to compare more fairly with other baseline models, we conduct experiments in other domains besides ani-

mal and plant.