# OpenReview forum: "Descriptive Prompt Paraphrasing for Target-Oriented Multimodal Sentiment Classification"
_EMNLP/2023/Conference — EMNLP 2023 Findings_

### Official Review · Reviewer_5uYa · 2023-07-20

**Soundness:** 3

**Excitement:**

3: Ambivalent: It has merits (e.g., it reports state-of-the-art results, the idea is nice), but there are key weaknesses (e.g., it describes incremental work), and it can significantly benefit from another round of revision. However, I won't object to accepting it if my co-reviewers champion it.

**Missing References:**

1. As for baselines, apart from large VLMs, there is also a scarcity of models designed specifically for TMSC tasks, such as SMP [1].

2. In section 4.1, references to well-known models like BERT [2], RoBERTa [3], ResNet [4], and Transformer [5] are missing.

[1] Junjie Ye, Jie Zhou, Junfeng Tian, Rui Wang, Jingyi Zhou, Tao Gui, Qi Zhang, Xuanjing Huang: Sentiment-aware multimodal pre-training for multimodal sentiment analysis. Knowl. Based Syst. 258: 110021 (2022)

[2] Jacob Devlin, Ming-Wei Chang, Kenton Lee, Kristina Toutanova: BERT: Pre-training of Deep Bidirectional Transformers for Language Understanding. NAACL-HLT (1) 2019: 4171-4186

[3] Yinhan Liu, Myle Ott, Naman Goyal, Jingfei Du, Mandar Joshi, Danqi Chen, Omer Levy, Mike Lewis, Luke Zettlemoyer, Veselin Stoyanov: RoBERTa: A Robustly Optimized BERT Pretraining Approach. CoRR abs/1907.11692 (2019)

[4] Kaiming He, Xiangyu Zhang, Shaoqing Ren, Jian Sun: Deep Residual Learning for Image Recognition. CVPR 2016: 770-778

[5] Ashish Vaswani, Noam Shazeer, Niki Parmar, Jakob Uszkoreit, Llion Jones, Aidan N. Gomez, Lukasz Kaiser, Illia Polosukhin: Attention is All you Need. NIPS 2017: 5998-6008

**Paper Topic And Main Contributions:**

This paper introduces UnifiedTMSC, a model designed for TMSC tasks. The method involves constructing a hard prompt on the text modality through task paraphrasing, and a soft prompt on the image modality using image prefix tuning. These prompts are then fused through cross attention to generate the final representation and enable classification. Experimental results demonstrate that UnifiedTMSC outperforms other methods on four datasets, yielding the best results.

**Questions For The Authors:**

A. Why does the structure for sentiment classification differ between the training and inference stages? Is this difference based on experimental comparisons?

B. Has the choice of setting the number of layers for cross-attention to 4 been experimentally validated, or is it subjectively determined? Moreover, does directly using a 12-layer Transformer have any impact?

C. It might be worth considering the utilization of LLMs for generating paraphrased prompts and conducting comprehensive tests to assess the robustness of UnifiedTMSC.

**Reasons To Accept:**

1. In comparison to several existing non-large language models, the method proposed in this paper attains superior results across four datasets.

2. The paper not only compares the performance of its own model with other approaches but also examines various text paraphrased forms, enhancing the comprehensiveness of the experiment.

3. In general, the paper's presentation flows smoothly, and the writing is clear.

**Reasons To Reject:**

1. The experiments do not encompass several recent methodologies involving large vision-language models (VLMs), such as Flamingo/OpenFlamingo and InstructBLIP (although relatively recent, it would be valuable to include them), which have demonstrated impressive zero/few-shot in-context learning performances on numerous vision-language tasks, including visual QA—tasks that the TMSC problem can be framed as.

2. The primary innovation of the paper lies in the deliberate design of task-related paraphrased prompts, while other works primarily combine existing methods. In this respect, the innovation presented in this paper appears to be somewhat limited.

3. The paper lacks a clear demonstration of the usefulness of the mentioned task description and seed prompt for subsequent paraphrased prompt design. As it stands, these two steps appear redundant.

4. The paper lacks essential experimental details, such as whether the Transformer used for cross-attention loads pre-training parameters, and the number of paraphrased prompts utilized for each data point during training.


**Reproducibility:**

4: Could mostly reproduce the results, but there may be some variation because of sample variance or minor variations in their interpretation of the protocol or method.

**Reviewer Confidence:**

4: Quite sure. I tried to check the important points carefully. It's unlikely, though conceivable, that I missed something that should affect my ratings.

---

> ### Author Rebuttal · Authors · 2023-08-28
>
> - **C1**: The experiments do not encompass several recent methodologies involving large vision-language models (VLMs), such as Flamingo/OpenFlamingo and InstructBLIP (although relatively recent, it would be valuable to include them), which have demonstrated impressive zero/few-shot in-context learning performances on numerous vision-language tasks, including visual QA—tasks that the TMSC problem can be framed as.
> - **R1**: Thank you for your careful review. We also believe that zero/few shot in-context learning is a challenging direction. Thank you for your suggestions and guidance, and we will continue to try and research in this area.
>
> - **C2**: The primary innovation of the paper lies in the deliberate design of task-related paraphrased prompts, while other works primarily combine existing methods. In this respect, the innovation presented in this paper appears to be somewhat limited.
> - **R2**: Thanks for your serious review. We transform the original aspect/entity level multimodal sentiment classification into the MLM pre-training task by adding prefix vectors and paraphrased prompts to the image-text data, which changed the task format. Therefore, it is crucial to design paraphrased prompts that are closer to the original MSC task, which is also our main innovation: **designing paraphrased prompts that are in line with the task under the prompt paradigm**.
>
> - **C3**: The paper lacks a clear demonstration of the usefulness of the mentioned task description and seed prompt for subsequent paraphrased prompt design. As it stands, these two steps appear redundant.
> - **R3**: Thanks for your comment. We are sorry that we don’t explain this clearly. Since we converted the original sentiment classification task into the MLM pre-trained task through adding prompts, it is crucial to design prompts that fit the original task. **As shown in the left part of Figure 2**, we obtain seed prompt and keyword from the task description. Subsequently, the seed prompt is transformed through the paraphrasing rule, and guide the generation of paraphrased prompts that are close to the task description. We will add a more detailed explanation in section 3.3.
>
> - **C4**: The paper lacks essential experimental details, such as whether the Transformer used for cross-attention loads pre-training parameters, and the number of paraphrased prompts utilized for each data point during training.
> - **R4**: Thank you for carefully reading and pointing out these shortcomings. We will supplement these details in the implementation details section. The pre-training parameters of Transformer for cross-attention are not loaded. For the Twitter2015 and Twitter2017, we use three paraphrased prompts P1, P2  and P3 obtained based on the paraphrasing rule, and compare them with three arbitrary prompts P1’ , P2’ and P3’. The results are reported in Table 6. The results indicate that the effect of paraphrased prompts is superior to the traditional prompts, and P1, P2  and P3 are comparable. In subsequent task, one of the paraphrased prompts can be selected for experimentation. P1, P2 , P3  P1’ , P2’ and P3’ are follows:
> “P1: {target} express a [MASK] sentiment.”,
> “P2: The emotion of {target} is [MASK].”,
> “P3: A [MASK] sentiment is expresses towards to {target}”.
> “P1’: I feel the {target} is [MASK].”,
> “P2’: The {target} made me feel [MASK].”
> “P3’: I [MASK] the {target}.”
> In addition, when the prediction label is a real value, we change the prompt P2 to “The score of the {target} is [MASK].” for the Multi-ZOL dataset. For the MASAD dataset, P2 is selected for experimentation.
>
>
> - **C5**: Why does the structure for sentiment classification differ between the training and inference stages? Is this difference based on experimental comparisons?
> - **R5**: Thank you for your comment. We apologize for any errors in our expression here. Our training process and inference stage are consistent.
>
>
> - **C6**:  Has the choice of setting the number of layers for cross-attention to 4 been experimentally validated, or is it subjectively determined? Moreover, does directly using a 12-layer Transformer have any impact?
> - **R6**: Thanks for your comment. During our experiment, we set the cross-attention layers to 4, 6, and 12, respectively. We find that setting to 4 has the best effect, while setting the cross-attention layers to 6 and 12 resulted in poorer experimental results. Taking the Twitter-2015 dataset as an example, the following are the experimental results when the number of cross-attention layers is set to 4, 6, and 12 (the paraphrased prompt used is “P1: {target} express a [MASK] sentiment.”)
> Twitter2015: (4 layers, Acc: 80.2, Macro-F1: 77.6), (6 layers, Acc:79.6 Macro-F1:77.2), (12 layers, Acc: 80.1, Macro-F1:77.4)
>
> - **C7**: It might be worth considering the utilization of LLMs for generating paraphrased prompts and conducting comprehensive tests to assess the robustness of UnifiedTMSC.
> - **R7**: Thank you for your suggestion. Our paraphrasing rule is designed based on the relative position and synonym substitution, which essentially requires manual experience. One of our future works is to use LLMs to automatically generate paraphrased prompts and avoid relying on human labor.
>
> - **C8**: Missing References
> - **R8**: Thank you for carefully reviewing our paper and identifying the issues. We will supplement some missing references based on your suggestion. We started **training from scratch** and only used datasets from related tasks for training and inference. SMP [1] is a multimodal pre training framework for multimodal sentiment analysis, which uses large image-text dataset for pre-training tasks and then performs fine-tuning in downstream tasks.
>
> [1] Junjie Ye, Jie Zhou, Junfeng Tian, Rui Wang, Jingyi Zhou, Tao Gui, Qi Zhang, Xuanjing Huang: Sentiment-aware multimodal pre-training for multimodal sentiment analysis. Knowl. Based Syst. 258: 110021 (2022)

---

### Official Review · Reviewer_KMtj · 2023-08-01

**Soundness:** 3

**Excitement:**

3: Ambivalent: It has merits (e.g., it reports state-of-the-art results, the idea is nice), but there are key weaknesses (e.g., it describes incremental work), and it can significantly benefit from another round of revision. However, I won't object to accepting it if my co-reviewers champion it.

**Missing References:**

[1] Seoh R, Birle I, Tak M, et al. Open Aspect Target Sentiment Classification with Natural Language Prompts[C]//Proceedings of the 2021 Conference on Empirical Methods in Natural Language Processing. 2021: 6311-6322.
[2] Li G, Lin F, Chen W, et al. Prompt-Based Learning for Aspect-Level Sentiment Classification[C]//International Conference on Neural Information Processing. Cham: Springer International Publishing, 2022: 509-520.
[3] Li C, Gao F, Bu J, et al. Sentiprompt: Sentiment knowledge enhanced prompt-tuning for aspect-based sentiment analysis[J]. arXiv preprint arXiv:2109.08306, 2021.
[4] Yu Y, Zhang D. Few-shot multi-modal sentiment analysis with prompt-based vision-aware language modeling[C]//2022 IEEE International Conference on Multimedia and Expo (ICME). IEEE, 2022: 1-6.

**Paper Topic And Main Contributions:**

This paper proposes a unified TMSC model for aspect-based and target-based multimodal sentiment analysis tasks. They model these two tasks via descriptive prompt paraphrasing based on language model. Moreover, they conduct extensive experiments on four TMSC datasets. The results show their model outperform the existing baselines.

**Questions For The Authors:**

1) Why do the authers use chinese-roberta-wwm-ext as textencoder for Multi-ZOL? How about the performance of BERT? The baselines should have the same base model for fair comparsion.
2) Why can UnifiedTMSC improve the performance of TMSC. The authors use a unified model to model two subtasks. However, there is no module to improve the performance for TMSC and these two tasks are trained independently.

**Reasons To Accept:**

1) The author propose a UnifiedTMSC model to reformulate aspect-based and target-based sentiment analysis.
2) This paper is well written and easy to follow.
3) Series of experiments indicate the effectiveness of their model.

**Reasons To Reject:**

1) The novelty of the proposed model is limited. The authors use discrete and continuous prompts to fine the language model, which is widely used in language model for aspect-based sentiment analysis [1-3] and TMSC [4]. The author should cite these references and compare with them.


**Reproducibility:**

4: Could mostly reproduce the results, but there may be some variation because of sample variance or minor variations in their interpretation of the protocol or method.

**Reviewer Confidence:**

5: Positive that my evaluation is correct. I read the paper very carefully and I am very familiar with related work.

---

> ### Author Rebuttal · Authors · 2023-08-28
>
> - **C1**: The novelty of the proposed model is limited. The authors use discrete and continuous prompts to fine the language model, which is widely used in language model for aspect-based sentiment analysis [1-3] and TMSC [4]. The author should cite these references and compare with them.
> - **R1**: Thank you for carefully reviewing our paper and identifying the issues. We will supplement some missing references based on your suggestion. Our data modality is different from that of the papers [1-3]. We will consider conducting experiments without the image modality and comparing these papers in the future. For paper [4], they concatenated image encoding and text prompt, and we use cross-attention to fuse them. We have tried to directly concatenate image and text features before, but this method did not work well on our datasets. We will add our discussion to the paper.
>
> - **C2**: Why do the authors use chinese-roberta-wwm-ext as text encoder for Multi-ZOL? How about the performance of BERT? The baselines should have the same base model for fair comparison.
> - **R2**: Thank you for pointing this out. For fair comparison, we re-conducted the experiment using BERT-base-chinese as the text encoder for Multi-ZOL, while we will modify the experimental results and implementation details. The experimental results using BERT-base-chinese as the text encoder on the Multi-ZOL dataset are as follows. In our experiment, only this dataset is Chinese, and the following results indicate that using the Roberta pre-training model is more effective.
> In our paper:  Multi-ZOL: (hf1/chinese-roberta-wwm-ext , Acc: 72.53, Macro-F1: 65.97)
> New Result:   Multi-ZOL: (BERT-base-chinese , Acc: 70.46, Macro-F1: 64.71)
>
> - **C3**: Why can UnifiedTMSC improve the performance of TMSC. The authors use a unified model to model two subtasks. However, there is no module to improve the performance for TMSC and these two tasks are trained independently.
> - **R3**: Thank you for carefully reviewing our paper and providing comment. We can conduct unified training and reference, or we can conduct training and reference separately. The former increases the amount of data to improve the effect, while the latter aims to compare the effects of different datasets on our model more fairly. For fairness in comparison with the baseline, we report the results of independent training for two tasks in the paper. Taking the Twitter-2015 and Twitter-2017 as the example, the experimental results of unified training and inference are as follows. (The paraphrased prompt used is “P1: {target} express a [MASK] sentiment.”)
> Twitter2015: (Acc: 81.3, Macro-F1: 79.8)
> Twitter2017: (Acc: 78.1, Macro-F1: 77.8)
> The results of separate training are:
> Twitter2015: (Acc: 80.2, Macro-F1: 77.6)
> Twitter2017: (Acc: 76.9, Macro-F1: 76.7)
>
> [1] Seoh R, Birle I, Tak M, et al. Open Aspect Target Sentiment Classification with Natural Language Prompts[C]//Proceedings of the 2021 Conference on Empirical Methods in Natural Language Processing. 2021: 6311-6322.
> [2] Li G, Lin F, Chen W, et al. Prompt-Based Learning for Aspect-Level Sentiment Classification[C]//International Conference on Neural Information Processing. Cham: Springer International Publishing, 2022: 509-520.
> [3] Li C, Gao F, Bu J, et al. Sentiprompt: Sentiment knowledge enhanced prompt-tuning for aspect-based sentiment analysis[J]. arXiv preprint arXiv:2109.08306, 2021.
> [4] Yu Y, Zhang D. Few-shot multi-modal sentiment analysis with prompt-based vision-aware language modeling[C]//2022 IEEE International Conference on Multimedia and Expo (ICME). IEEE, 2022: 1-6.

---

### Official Review · Reviewer_xjQS · 2023-08-07

**Soundness:** 3

**Excitement:**

4: Strong: This paper deepens the understanding of some phenomenon or lowers the barriers to an existing research direction.

**Missing References:**

Hu, Guimin, et al. "UniMSE: Towards Unified Multimodal Sentiment Analysis and Emotion Recognition." Proceedings of the 2022 Conference on Empirical Methods in Natural Language Processing. 2022.

**Paper Topic And Main Contributions:**

This paper presents the UnifiedTMSC model for Target-Oriented Multimodal Sentiment Classification (TMSC) that leverages descriptive prompt paraphrasing to eliminate the division between entity-level and aspect-level tasks.
The model creates a unified task modelling framework using task paraphrasing and image prefix tuning, suggesting that sentiment polarity of the target is context-dependent rather than type-dependent. UnifiedTMSC demonstrates good performance on four diverse datasets, indicating its broad applicability and effectiveness.

**Questions For The Authors:**

1. The usage of synonyms is unclear. What does the synonym consist of? Are sentiment and emotion have their corresponding synonym dictionaries? Will these synonyms overlap?
2. Which prompt will the author use for the final inference?
3. Is all label only require one token to be generated? For example, does the vocabulary have a single token for the sentiment label "10"?
4. How do you deal with ill-format decode labels? Do you apply post-process normalization? For example, the ground truth is "positive", but the model generates "positively."
5. Significant tests are needed to prove the effectiveness, such as single-tail t-tests.

**Reasons To Accept:**

1. This proposes UnifiedTMSC uses a unique combination of prompt-based language modelling and descriptive prompt paraphrasing to achieve high performance on both entity-level and aspect-level tasks.
2. The rigorous experiments conducted on four different datasets lend credibility to the approach and suggest it could be a promising avenue for future research in TMSC.

**Reasons To Reject:**

1. Novelty is somehow limited and more like a technical report; rule-based prompt data enhancement is the main innovation.
2. It does not follow the pattern of unified training followed by direct inference across multiple tasks and datasets. (without dataset-specific fine-tuning)
3. Important details are unclear and should be clarified. The description of the model's inner workings and implementation details could be more elaborate for the readers to replicate the study.

**Reproducibility:**

4: Could mostly reproduce the results, but there may be some variation because of sample variance or minor variations in their interpretation of the protocol or method.

**Reviewer Confidence:**

3: Pretty sure, but there's a chance I missed something. Although I have a good feel for this area in general, I did not carefully check the paper's details, e.g., the math, experimental design, or novelty.

**Typos Grammar Style And Presentation Improvements:**

* The overall writing could be improved in terms of clarity and formality.
* Adding the decoding example in Figure 2 is recommended for better understanding.
* Formula (4) is quite confusing; maybe it could be omitted.
* Please provide definitions for variables at their first point of use. For instance, the text embedding 'R' is initially mentioned in line 212 but is not clearly defined until line 308. This sequence could confuse readers and disrupt the flow of the paper. Please consider revising for clarity.
* Line 297: spliced —> concat?

---

> ### Author Rebuttal · Authors · 2023-08-28
>
> - **C1**: Novelty is somehow limited and more like a technical report; rule-based prompt data enhancement is the main innovation.
> - **R1**: Thanks for your serious comment. Our work has improved by 1.5%, 1.7%, 3.77%, and 2.19% on Macro-F1 on the four datasets of Twitter-2015, Twitter-2017, Multi-ZOL, and MASAD, respectively. In addition, our work has transformed the original classification task into the MLM pre-training task by adding prefix vectors and paraphrased prompts to the image-text data, which has changed the task format. Therefore, it is crucial to design paraphrased prompts that are closer to the multimodal sentiment prediction task, which is also **our main innovation: designing paraphrased prompts that are in line with the task description under the prompt paradigm**.
>
> - **C2**: It does not follow the pattern of unified training followed by direct inference across multiple tasks and datasets. (without dataset-specific fine-tuning)
> - **R2**: Thank you for reading our paper carefully and providing comments. For fair experimental comparison, each task is trained on their own dataset. If we conduct unified training and inference for multiple datasets, it increases the amount of training data. We add some new experimental results by training these tasks together. Taking the Twitter-2015 and Twitter-2017 as the example, the experimental results of unified training and inference are as follows. (The paraphrased prompt used is “P1: {target} express a [MASK] sentiment.”)
> Twitter2015: (Acc: 81.3, Macro-F1: 79.8)
> Twitter2017: (Acc: 78.1, Macro-F1: 77.8)
> The results of separate training are:
> Twitter2015: (Acc: 80.2, Macro-F1: 77.6)
> Twitter2017: (Acc: 76.9, Macro-F1: 76.7)
>
> - **C3**: Important details are unclear and should be clarified. The description of the model's inner workings and implementation details could be more elaborate for the readers to replicate the study.
> - **R3**: Thank you for your suggestion. We will supplement the experimental details and open our code in GitHub. About the paraphrased prompt for each task, we design three paraphrased prompts P1, P2  and P3 . In addition, three traditional prompts P1’ , P2’ and P3’ are used to compare with paraphrased prompts. The results are reported in Table 6. The results indicate that the effect of paraphrased prompts is superior to the traditional prompts, and P1, P2  and P3 are comparable. In subsequent task, one of the paraphrased prompts can be selected for experimentation. P1, P2 , P3  P1’ , P2’ and P3’ are follows:
> “P1: {target} express a [MASK] sentiment.”,
> “P2: The emotion of {target} is [MASK].”,
> “P3: A [MASK] sentiment is expresses towards to {target}”.
> “P1’: I feel the {target} is [MASK].”,
> “P2’: The {target} made me feel [MASK].”
> “P3’: I [MASK] the {target}.”
> In addition, when the prediction label is a real value, we change the prompt P2 to “The score of the {target} is [MASK].” for the Multi-ZOL dataset. For the MASAD dataset, P2 is selected for experimentation.
>
> - **C4**: The usage of synonyms is unclear. What does the synonym consist of? Are sentiment and emotion have their corresponding synonym dictionaries? Will these synonyms overlap?
> - **R4**: Thank you for your comment. We‘re sorry that we don’t explain the synonyms clearly. In **Bing Dictionary**, when the meaning of 'sentiment' is 'a feeling or emotion', the similar words are 'feeling’ and 'emotion' respectively. When the meaning of 'sentient' is 'a view of or attitude towards a situation or event; an opinion', the similar words of ‘sentiment’ are such as ‘view, feeling, opinion’, etc. These similar words form a dictionary of synonyms, and we take the word from this dictionary to replace the ‘sentient’ of the seed prompt.
>
> - **C5**: Which prompt will the author use for the final inference?
> - **R5**: Thank you for pointing this out. Refer to R3, we design three different paraphrased prompts from traditional prompts, and the effects of these three paraphrased prompts are **comparable**. Therefore, we can choose any one of them for training and inference in the experiment. For example, for the MASAD dataset, we use P2 to conduct experiment.
>
> - **C6**: Is all label only require one token to be generated? For example, does the vocabulary have a single token for the sentiment label "10"?
> - **R6**: Thanks for the review. We ‘re sorry that we didn't explain it clearly. In our actual experiment, we convert digital sentiment labels into Chinese characters, such as "10" to "ten", and the converted Chinese characters correspond to a single token in the vocabulary. We will supplement it in A.2 of Appendix.
>
> - **C7**: How do you deal with ill-format decode labels? Do you apply post-process normalization? For example, the ground truth is "positive", but the model generates "positively."
> - **R7**: Thank you for your comment. In the inference stage, if non-label words are generated, we use the **answer engineering** in prompt to map the non-label words to label words. For example, when we use “P1’: I feel the {target} is [MASK].” for training and prediction, the words generated by the model are “good”, “ok”, and “bad”. Therefore, we need to use answer engineering to map these words to label words “positive”, “neutral” and “negative”. We notice that the **autoregressive** model XLNet can alleviate the problem of generating non-label words, and our future work will consider this.
>
> - **C8**: Significant tests are needed to prove the effectiveness, such as single-tail t-tests.
> - **R8**: Thank you for your suggestion. We will conduct the single-tail t-tests to verify the effectiveness of our model and supplement the results in the experimental results tables. Taking the Multi-ZOL dataset as an example, the P-Values between our model and the baselines are as follows:
> MIMN ：0.0316
> ModalNet:  0.0266
> CLIP:    0.0092
>
> - **C9**: Typos Grammar Style And Presentation Improvements:
> - **R9**: Thank you for carefully reading our paper and pointing out these issues. We will make revisions to the paper based on your suggestions and we will adopt **professional proof-reading service**.

---

### Meta-Review · Area_Chair_LdN1 · 2023-09-25

**Recommendation:** 4

**Metareview:**

Based on the idea that sentiment polarity of the specific target is not governed by its type but its context, authors propose a unified model for target-oriented multimodal sentiment classification, UnifiedTMSC. It is prompt-based language modelling and performers well on four datasets over two target types.

Reviewers give all good (3) in soundness; two ambivalent (3) and one strong (4) in excitement. Reviewers have some concerns about novelty (the use of prompts for language modeling is somehow not so exciting), clarity of details and reproducibility. However, this paper is well written and the comprehensive experiments support good experimental results.

---

### Decision · Program_Chairs · 2023-10-07

**Decision:**

Accept-Findings

**Comment:**

Based on the idea that sentiment polarity of the specific target is not governed by its type but its context, authors propose a unified model for target-oriented multimodal sentiment classification, UnifiedTMSC. It is prompt-based language modelling and performers well on four datasets over two target types.

Reviewers give all good (3) in soundness; two ambivalent (3) and one strong (4) in excitement. Reviewers have some concerns about novelty (the use of prompts for language modeling is somehow not so exciting), clarity of details and reproducibility. However, this paper is well written and the comprehensive experiments support good experimental results.